# Swedish Trotting Horse Trainers’ Perceptions of Animal Welfare Inspections from Public and Private Actors

**DOI:** 10.3390/ani12111441

**Published:** 2022-06-03

**Authors:** Frida Lundmark Hedman, Ivana Rodriguez Ewerlöf, Jenny Frössling, Charlotte Berg

**Affiliations:** 1Department of Animal Environment and Health, Swedish University of Agricultural Sciences, P.O. Box 234, SE-532 23 Skara, Sweden; jenny.frossling@sva.se (J.F.); lotta.berg@slu.se (C.B.); 2Department of Disease Control and Epidemiology, National Veterinary Institute (SVA), SE-751 89 Uppsala, Sweden; ivana.ewerlof@sva.se

**Keywords:** compliance, control, experience, legislation, private standards

## Abstract

**Simple Summary:**

Harness racing is the most common form of horse racing in Sweden. As public awareness of animal welfare is increasing, the welfare of these horses must be ensured. Trotting horse trainers in Sweden undergo an official animal welfare inspection by the County Administrative Board (CAB) and a private inspection by their own association, the Swedish Trotting Association (STA). This study investigated trainers’ perceptions of these different inspections using a digital questionnaire sent out during spring 2021. Of the 396 responding trainers, a majority reported quite positive experiences of both CAB and STA inspections. However, most perceived the STA inspections to be more valuable and the STA inspectors to be more competent than the CAB inspectors. Overall, the competence and manner of the inspector had a stronger association with trainers’ perceptions of an inspection than the results of the inspection. While trainers were generally satisfied with the control system, they would like better coordination between the different inspections.

**Abstract:**

In Sweden, the County Administrative Board (CAB) and Swedish Trotting Association (STA) both perform animal welfare inspections of the premises of trotting horse trainers. The CAB inspection checks for compliance with the legislation, and the STA inspection checks for compliance with the private ‘Trotter Health Standard’, which mainly sets the same requirements as the legislation. This study investigated the views of trainers on these inspections both as separate events and in relation to each other. A digital questionnaire was sent out to trotting horse trainers in Sweden during spring 2021, and 396 trainers responded. Descriptive and statistical analyses were used to evaluate the responses. In general, the trainers reported positive experiences of both the CAB and STA inspections, but they had consistently more positive views about the private STA inspections than the official CAB inspections. The outcome of the inspections, i.e., non-compliance or not, did not affect trainers’ perceptions of the inspections, but inspectors’ knowledge, manner, and responsiveness had a strong effect. The trainers were generally satisfied with the current control system but would like better coordination between the different inspections.

## 1. Introduction

Public awareness of animal welfare has increased in recent decades [1], as reflected in increased policy activity [2]. The European Union (EU) has established animal welfare legislation for farm animals and laboratory animals, and several countries both inside and outside the EU have developed national legislation and policies to protect the welfare of animals [3,4]. In addition to legislation, use of private animal welfare standards, initiated by private actors wishing to quality assure their activities, is increasing [5,6,7,8]. However, current legislation and standards mainly cover rearing and management of production animals in order to ensure quality and attract consumers of meat and dairy products [9]. There is no common EU legislation on the keeping, management, and use of horses, except during transport [10] and slaughter, although the horse racing industry is regularly the subject of public debate. With increasing public understanding and interest in animal welfare, conditions in the racing industry are being questioned [11,12,13], and industry initiatives to assure and improve racehorse welfare have been initiated, e.g., for thoroughbred horses in New Zealand [14].

In Sweden, harness racing (trotting horses) is the most common form of horse racing, with 8200 races annually and approximately 16,000 horses trained by approximately 400 professional trainers (A license) and 3300 amateur trainers (B license) [15]. All horses in Sweden are protected by national legislation, i.e., the Animal Welfare Act (SFS 2018:1192), the Animal Welfare Ordinance (SFS 2019:66), and the Swedish Board of Agriculture’s regulations and general advice on the keeping of horses (SJVFS 2019:17) and on training and competition with animals (SJVFS 2019:26). The County Administrative Boards (CABs), i.e., the regional, competent authorities in Sweden since 2009, are responsible for monitoring compliance with the legislation [16]. However, due to the relatively low frequency of CAB inspections and a need to improve audits made by the local licensing committees situated at Swedish racetracks, the Swedish Trotting Association (STA) introduced a private standard on animal welfare, called the ‘Trotter Health Standard’ [17,18]. The aim of the standard is to ensure horse welfare, to demonstrate and assure the quality of the sport and management of the horses, and to increase trotting horse trainers’ knowledge and awareness of animal welfare [17]. Both professional and amateur trainers must comply with this standard in order to keep their license. One aspect of the standard is that inspectors (internally called ‘controller’ or ‘auditor’ by the organization but here referred to as inspectors), hired by the STA, make inspections (internally called ‘audits’) on each trainer’s horse premises, normally every four years, to check for compliance with the standard. These inspections by STA have been running since 2015.

The Trotter Health Standard is mainly on the same level as the Swedish animal welfare legislation but with some additional requirements relating to equipment, safety, and medical treatments. Thus trotting horse trainers in Sweden have both official and private inspections carried out on their premises, checking for compliance with mainly the same requirements. In previous studies, we found that how compliance is measured differs between sets of regulations based on the same requirements [19,20]. A resource-based requirement can, for example, be measured using different combinations of resource-, management-, and animal-based measures, and the animals’ welfare states can be measured either on an individual or a group level, leading to different outcomes of the inspections. In the present study, we examined how trotting horse trainers perceive these two different inspections, i.e., the official CAB inspection and the private STA inspection. We hypothesized that perceptions may differ between having an official and private inspection and that demographic factors, as well as other factors, may influence these perceptions.

The specific objectives of the study were to investigate how trotting horse trainers in Sweden perceive the official animal welfare inspections and the STA private audits, both separately and in relation to each other, and to identify any factors that potentially influence their perceptions.

## 2. Material and Methods

### 2.1. Questionnaire

An electronic questionnaire (see Appendix A) asking questions about trotting horse trainers’ experiences and expectations related to animal welfare inspections from CAB and STA was developed in the software program Netigate (version 8). The questionnaire was validated stepwise by performing alternating test runs and making improvements before it was finalized. Email addresses of A- and B-licensed trainers in Sweden were obtained from STA and entered in Netigate. Of a total of 3496 trainers in Sweden, an email address was obtained for 2896 trainers. A link to the electronic questionnaire was sent to these trainers from Netigate in March 2021. The trainers were given six weeks to respond and were reminded twice. The data received were analyzed anonymously. The study and questionnaire were approved by the Swedish Ethical Review Authority (reference number: Dnr 2019–06370).

The questionnaire consisted of four parts: (1) Information and background on the respondent and their trotting horse business and their thoughts on animal welfare and its importance; (2) respondents’ views on getting both CAB and STA inspections; (3) respondents’ experiences and expectations on animal welfare legislation and the official inspections; and (4) respondents’ experiences and expectations on the Trotter Health Standard and the STA inspections. The questionnaire consisted of 77 questions in total, but respondents did not have to answer all of these. For example, trainers that had received an inspection more than three years previously were asked questions about their expectations for the next inspection, while trainers that had received an inspection within the previous three years were asked questions relating to their experiences of this inspection. The questions were mainly of the closed type. The respondents were asked to choose from a list of options or state their opinion on a five- or 10-point Likert scale (from 1 = fully disagree to 5 or 10 = fully agree). There were also a couple of open-ended questions where the respondent could clarify the answer or express an opinion without being given any options to choose between.

### 2.2. Statistical Analysis

#### 2.2.1. Data Preparation

All responses from the questionnaire (except from open-ended questions) were summarized descriptively and visualized using Netigate (version 8). This summary was assessed to get an initial overview of all responses. For further statistical analysis, data preparation was carried out.

For categorical questions about the age, education, working experience, physical and mental health of the respondent, and about whether the inspection was announced beforehand, the responses were grouped into fewer groups based on number of replies in each original category. Replies indicating the geographical location of the stables were based on the 21 counties of Sweden, i.e., level 3 of the Nomenclature of Territorial Units for Statistics (NUTS). These were grouped into three larger regions, NUTS level 1 (SE1—East Sweden, SE2—South Sweden, and SE3—North Sweden). The gender category ‘Other/do not want to respond’ was removed and the replies treated as missing. Replies to the question about the age and sex of the inspector/s compared with the respondent were grouped into two categories: ‘Only younger woman/women’ and ‘Remaining ages and sexes’. Detailed information on how the responses were regrouped can be seen in the Appendix A. The response category ‘Do not remember’ was removed, and the replies treated as missing.

Depending on the type of statistical analysis, questions with numerical graded responses were either kept numerical, and the answer ‘Don’t know’ was removed, or categorized into intervals of the grading, and the answer ‘Don’t know’ was kept. For the latter, questions graded 1–5 were grouped into the three categories 1–2, 3, and 4–5, and questions graded 1–10 were grouped into the three categories 1–3, 4–7, and 8–10.

#### 2.2.2. Statistical Tests and Analyses

Data were prepared and analyzed using R software [21]. Spearman’s correlation was used to investigate the relationship between the replies to numerical questions. Spearman’s correlation coefficient *r* takes a value between −1 and 1, where 0 indicates no correlation and −1 or 1 indicates a perfect correlation between the variables. Associations between replies to categorical or categorized questions were tested with Pearson’s chi-square test or Fisher’s exact test. The statistical methods were chosen with consideration given to the limited number of responses and the desire not to complicate the analysis more than necessary to achieve the desired purpose.

Questions regarding demographics, understanding and expectations on inspections, inspector traits and fair treatment, and the outcomes of the inspections were tested against trainers’ perceptions of CAB and STA inspections, respectively. Corresponding questions about CAB and STA inspections were also compared with each other to investigate potential differences. To enable comparison with results presented by Väärikkälä and co-authors [22], the association between having been inspected or not and the perceived necessity of the inspections was also investigated.

## 3. Results

### 3.1. Demographics and Information about Respondents

Of the 2896 trainers that received a link to the questionnaire, 396 responded to some extent. Of these, 248 submitted complete answers, i.e., answered all intended questions. Trainers from all 21 counties in Sweden participated.

A majority of the respondents (63%, 252/396) were above 50 years of age, and most were amateur trainers (B license) with fewer than four horses in training (Figure 1). A majority of the trainers (75%, 290/387) only trained their own horses, 23% (88/387) trained both their own and others’ horses, and 2% (9/387) only trained others’ horses. Most of the respondents stated that they had the trotting horse business mainly as a hobby (64%, 253/391), while 10% (39/391) stated that the horse business was their only income. Only 6% (23/391) of the trainers had employees, while 94% (368/391) did not. Of the respondents, 71% (274/385) had their own premises, while 29% (111/385) shared stables with other trainers. Most trainers kept their horses in single boxes (92%, 355/385), 37% (143/385) kept horses loose-housed (i.e., in open barns), and only 1% (2/385) kept their horses tied in stalls. Most trainers (97%, 370/381) described conditions in their stable as good to very good. The trainers who completed the questionnaire really seemed to enjoy training horses, reporting a mean value of 9.48 on a scale from 1 (I certainly do not) to 10 (I certainly do).

Almost half of the trainers (48%, 145/299) had received an official CAB inspection since 2009. Of these, 48% (70/145) responded that they had received a CAB animal welfare inspection within the past three years. The corresponding proportion that had received an audit from the STA within the past three years was 83% (214/257).

### 3.2. Respondents’ General Views on Animal Welfare and Regulations

When asked about the three most important factors for good animal welfare (in addition to good health, feed, and water), 53% (204/382) of the trainers chose the option ‘quick treatment of sick or injured horses’, 52% (197/382) indicated ‘daily exercise in outdoor paddocks’, and 48% (184/382) indicated ‘good care and management of the horse’. The three factors selected fewest times were: ‘access to summer pasture’ (2%, 8/382), ‘plenty of space in the stable’ (5%, 21/382), and ‘keeping the horses loose-housed’ (6%, 24/382). Only 5% (18/351) of the trainers stated that they sometimes had to deprioritize the welfare of their horses in favor of other business. A majority (88%, 311/351) responded that they always prioritize horse welfare, and 78% (260/337) had not felt any financial pressure that would compromise horse welfare. Still, 39% (142/364) responded that they would like to give higher priority to horse welfare.

In general, 63% (204/325) of the trainers were satisfied with the Swedish animal welfare legislation, and 83% (196/236) were satisfied with the Trotter Health Standard. Most trainers reported that it is easy to know what is expected in order to comply with the legislation (73%, 238/326) and the Trotting Health Standard (85%, 205/241). Twelve percent of respondents (38/320) believed that the legislation consists of complicated requirements that can be difficult to comply with, and 8% (20/262) believed that the Trotting Health Standard sets complicated requirements. However, 44% (140/320) and 60% (156/262), respectively, were not of the opinion that the legislation or Trotting Health Standard sets any complicated requirements. For this question, 44% (142/320) selected the option ‘Do not know’ (i.e., they could not answer the question) regarding the current legislation, and 33% (86/262) ticked ‘Do not know’ regarding the Trotting Health Standard. Of the trainers surveyed, 23% (74/315) agreed with the statement that the legislation only consists of requirements relevant for horse welfare, while the corresponding proportion for the Trotter Health Standard was 51% (134/262). However, 18% (56/315) believed that there are some requirements in the legislation that are irrelevant for horse welfare, and 6% (17/262) believed that the Trotting Health Standard includes irrelevant requirements. Note that many trainers could not answer this question, since 59% (185/315) ticked ‘Do not know’ for the legislation, and 42% (111/262) ticked ‘Do not know’ for the Trotter Health Standard.

The main reasons stated by the trainers for complying with the regulations were that: they wanted their horses to be well (legislation: 93%, 304/330; Trotter Health Standard: 95%, 246/259), they wanted to contribute to the trustworthiness of Swedish harness racing (legislation: 84%, 275/326; Trotter Health Standard: 88%, 213/241), and they believed that a serious business must comply with the regulations (legislation: 83%, 272/330; Trotter Health Standard: 95%, 230/243). The trainers agreed (95%, 311/328) on the importance of good animal welfare for the reputation of the Swedish trotting horse business, with 92% (299/324) responding that trainers who constantly violate the animal welfare legislation destroy trust in the trotting horse business.

### 3.3. Respondents’ General Views on Inspections

A majority of the trainers, 55% (165/303), agreed that the official CAB control is necessary in order to check that animal welfare on trotting horse premises is satisfactory. However, a greater proportion (85%, 221/258) viewed the STA inspections as necessary for animal welfare, and 89% (230/259) agreed with the statement that ‘I believe it is good that the Trotting Association makes inspections at my place’. The number of trainers agreeing with the same statement for CAB inspections was 61% (184/303).

Most trainers (82%, 288/350) agreed that receiving an inspection at least every three years from either the CAB or STA is reasonable, with only 3% (10/350) preferring an inspection rate exceeding once every five years. However, the trainers had varying opinions on whether it is necessary to have both official and private inspections concerning animal welfare for trotting horses. Of the respondents, 41% (143/351) stated that these double inspections are unnecessary and 40% (140/351) that they are necessary. A majority of the trainers (61%, 215/350) would appreciate better coordination between the CAB and STA, although 46% (160/350) claimed to be satisfied with the current system. Only 18% (63/350) were not satisfied with the current situation. Eleven percent (39/351) of the trainers disagreed with the statement that CAB and STA inspectors perform the same assessments on their premises, 28% (97/351) did not know, and 44% (156/351) agreed with the statement that the same assessment is made, i.e., the inspection results are the same in both a CAB and STA inspection. Some trainers (12%, 42/350) stated that it could be difficult to differentiate between the inspection types, while most (58%, 201/350) trainers replied it was easy to keep the two different types of inspections apart. Of the trainers, 43% (158/351) responded that they know the similarities and differences regarding requirements and assessments between the legislation and the Trotter Health Standard. However, some trainers disagreed with the statement that they are knowledgeable about similarities and differences (10%, 37/351) or indicated that they did not know (18%, 64/351).

The trainers were also asked about what they would do if they wanted information about the animal welfare inspection (CAB or STA). A majority of the trainers (71%, 242/340) responded that they would turn directly to the CAB or STA, depending on the type of inspection, and 48% (163/340) said that they would contact the individual inspector conducting the inspection. However, 57% (193/340) reported that they would ask the STA, irrespective of the type of inspection. The trotting horse trainers relied to a smaller extent on information from other trainers (25%, 84/340), their veterinarian (20%, 67/340), or social media (13%, 43/340).

### 3.4. Respondents’ General Views on Inspector Traits

Being knowledgeable about horses and the trotting horse business was the most important characteristic of an animal welfare inspector according to the trainers (52%, 178/345). Knowledge about animal welfare in general was the second most important trait (40%, 139/345). Four different characteristics came third and had approximately the same importance. These were: ‘the inspector can make uniform assessments’ (30%, 104/345), ‘the inspector is smooth and can make flexible assessments’ (30%, 102/345), ‘the inspector can justify non-compliance so the trainer understands why it is important to take measures’ (30%, 102/345), and ‘the inspector is knowledgeable about the regulations on which their inspection is based’ (29%, 99/345). The inspector traits that seemed to be least important were: ‘the inspector is a good listener’ (1%, 5/345), ‘the inspector shows understanding that as a horse trainer I am under financial and time pressure’ (1%, 4/345), and ‘the inspector is knowledgeable about the administrative procedures and processing of matters/cases’ (1%, 4/345).

### 3.5. Perceptions on Being Inspected

The trainers that had received an inspection from the CAB and/or an audit from the STA within the past three years reported a positive experience in general (Figure 2). However, they reported a significantly more positive experience (Fisher’s test, *p* < 0.001) for the STA inspections than for the CAB inspections.

How the respondents graded their perception of the two inspections was significantly positively correlated (Spearman’s *r* = 0.42, *p* = 0.002, *n* = 54); i.e., if they were positive about the STA inspections, they were also positive about the CAB inspections (Table 1).

### 3.6. Factors Associated with Trainers’ Perceptions of an Inspection

#### 3.6.1. Demographics

Age, region of residence, educational level, working experience, health status, and type of training license (A or B) showed no association with how the trainers experienced their latest CAB or STA inspection. The only demographic factor that seemed to matter was the gender of the trainer for the STA inspection, where women were more positive (Fisher’s test, *p* = 0.007). There was no association between gender and the experience of a CAB inspection.

#### 3.6.2. Understanding and Expectation

The trainers’ understanding of the regulations and expectations before an inspection were, for some questions, associated or correlated with how the inspection was experienced. The satisfaction about the regulations, the contentment with the current situation (receiving both CAB and STA inspections), and the perception that the regulations only contain requirements relevant for horse welfare were associated or correlated with how an inspection was experienced for both the CAB and STA inspections. These results are shown in Table 2 and Table 3, where Table 2 shows results from correlation tests with numerical questions and mean response per question, and Table 3 shows results from tests on association for categorical questions, together with response distributions per category for each question. However, the trainers indicated that is easier to understand what is required of them in order to comply with the Trotter Health Standard than with the official legislation (chi-square test, *p* = 0.001).

A majority of the trainers replied that they were not worried at all before an inspection. On a scale of 1 (very worried) to 10 (not worried at all), 82% (57/70) responded 8–10 for the CAB inspections, and 84% (180/213) responded 8–10 for the STA inspections. A few trainers were worried before an inspection, with 7% (5/70) responding 1–3 for the CAB inspections and 2% (4/213) responding 1–3 for the STA inspections. However, there was no significant difference between the CAB or STA inspections regarding how worried the trainers were before an inspection (Fisher’s test, *p* = 0.062). Not being worried before the inspection was significantly positively correlated with the perception of an inspection from STA (but not from CAB), indicating that the trainers who were not worried (higher grading) had a better experience of the STA inspection, although the correlation was quite weak (Spearman *r* = 0.18). Few trainers prepared themselves or their stables before an inspection, and a majority (CAB: 89%, 190/213; STA: 86%, 60/70) did not prepare in any specific way before an inspection. The minority of trainers who reported preparing for an inspection stated that, e.g., they read the legislation/standard, looked up previous inspection results, or fixed things that they thought would be included in the inspection.

Whether the respondents had received an inspection or not was not significantly associated with their understanding of the necessity of the CAB or STA inspections. Most trainers (of those who had not had any recent inspection, i.e., during the past three years) believed that, if they were inspected tomorrow, the inspectors would not find many non-compliances. Actually, none (0%, 0/40) of the trainers fully agreed with the statement that the STA inspector would find non-compliances, and only 2% (5/218) felt certain that a CAB inspector would find non-compliances. Most of these trainers also expected the inspector to rate their horse keeping as good (CAB: 95%, 206/218; STA: 85%, 34/40).

#### 3.6.3. Inspector Traits and Fair Treatment

How a trainer perceived an inspector’s competence, manner, behavior, and responsiveness was important for how both a CAB and STA inspection was experienced (Table 2). A perception of being treated fairly, assessed in a correct and confident way, and given good advice on measures needed to reach compliance also affected the trainers’ experience of an inspection (CAB or STA) (Table 2). However, the STA inspectors were perceived as more competent (Fisher’s test, *p* < 0.001) and more interested in the trainer’s trotting horse business (chi-square test, *p* < 0.001). The trainers declared that, in general, they had a slightly better experience with inspectors from the STA than from the CAB (Table 2). However, 75% (49/66) perceived the CAB inspector to be pleasant with good intentions. The corresponding value for the STA inspector was 91% (193/211). Of the trainers, 64% (42/66) replied that the CAB inspector was knowledgeable and acted professionally, 53% (45/66) replied that the CAB inspector was interested in the trainer’s horse business, and 57% (38/66) replied that the CAB inspector was capable of explaining and motivating the assessments made. The corresponding values for the STA inspector were 88% (186/210), 85% (177/210), and 87% (183/210). Most of the trainers believed that the latest inspections from both the CAB (67%, 44/66) and the STA (92%, 195/212) had been fair, with only 2% (4/212) reporting unfairness in relation to an STA inspection and 18% (12/66) reporting unfairness in relation to a CAB inspection.

How a trainer perceived an inspection was not significantly associated with gender or age of the inspector/s (Table 3), but the mean experience was slightly lower if the inspector was a younger woman (both for CAB and STA). The CAB inspections were mostly performed by younger women (i.e., younger than the trainers) (Table 3). The CAB inspector was never a younger male, and, hence, it was not possible to analyze gender effects for young male CAB inspectors. In total, only 11% (7/62) of the CAB inspections were carried out by a male (either older or the same age as the trainer). The STA inspectors had a more diverse gender and age distribution. Most often, only one inspector was present during a CAB or STA inspection. However, it was more common for two inspectors to be present during a CAB inspection (Table 3). The number of inspectors present during an inspection was not associated with how the trainers perceived the inspection (Table 3).

The STA inspections were usually announced in advance, with only 5% (10/213) of STA inspections not being pre-announced. Of the CAB inspections, 26% (16/62) were carried out without any pre-announcement. Whether an inspection was pre-announced was not significantly associated with trainers’ perceptions of the CAB inspections (Table 3). The reason behind a CAB inspection was also not associated with trainers’ perceptions (Table 3), and, hence, it did not matter significantly whether the inspection was a planned, routine inspection initiated by CAB or an inspection initiated due to complaints from the public or whether it was the first inspection or an extra inspection due to previous non-compliances. The most common type of inspection was a planned, routine inspection for both the CAB and the STA (Table 3). For some of the trainers, the inspection was due to a complaint to the CAB that the trainer might have shortcomings in their horse-keeping system, and, for a few trainers, the CAB inspection was an extra inspection due to previous non-compliances (Table 3). Only one trainer had an extra inspection by the STA. Some trainers did not know the reason for the inspection (CAB: 14%, 10/69; STA: 7%, 15/213).

#### 3.6.4. Outcome of an inspection

Both CAB and STA inspectors found non-compliances during inspections. Despite this, no significant association was found between the trainers’ experience of inspections and detection of non-compliance at the time of inspection (*p* = 0.40 for CAB, *p* = 0.15 for STA) (Table 3). Of the responding trainers, approximately one-third reported that non-compliances were detected on their premises by both CAB and STA inspectors (Table 3). The most common types of non-compliances were related to the stable and interior design. One trainer had non-compliances related to horse management and welfare (during a CAB inspection). The most common way for both the CAB and the STA to handle non-compliance was to make a note about it in the inspection report. None of the trainers had received any harsher sanction from the STA, and only three of the trainers had received injunctions from the CAB. None of the respondents had been subjected to decisions on seizure of horses or the withdrawing of their license. The responding trainers indicated that the non-compliances noted by CAB inspectors were more expensive to rectify than the non-compliances found by the STA inspectors (significant difference, Fisher’s test, *p* = 0.04). The belief that it was more expensive to rectify CAB non-compliances was associated with a more negative perception of the inspection (Table 2). The general view of the trainers was that neither the CAB nor the STA is too rapid in taking strict actions and sanctions when someone does not comply with their requirements. On the contrary, more of the trainers seemed to think that the CAB (52%, 156/302) and the STA (57%, 146/258) are too slow to take strict actions, while 11% (34/302) believed that the CAB, in general, is too rapid in taking strict action and 4% (10/258) that the STA is too rapid in this regard.

A significantly larger proportion (Fisher’s test, *p* = 0.005) of trainers replied that STA inspectors gave them good advice on how compliance could be achieved (87%, 64/74) compared with the proportion reporting that CAB inspectors gave good advice (55%, 10/18). A majority of the trainers did not think that the CAB or STA inspection, in general, had led to an improvement in horse welfare on their premises (CAB: 79%, 52/66; STA 57% (120/211). Only a few trainers perceived that the inspection had improved horse welfare (CAB: 2%, 1/66; STA: 19% (40/211), although the grading was significantly higher for the STA inspection (chi-square test, *p* = 0.001). However, this was not significantly correlated with the perception of the inspection, i.e., a trainer could report a positive or negative experience regardless of whether the inspection was believed to improve animal welfare or not. After an inspection from either the CAB or the STA, most of the trainers perceived that the written inspection report reflected what had been said during the time of inspection (Table 3). This was significantly associated with the experience of the STA inspections but not with the CAB inspections. Most of the trainers (CAB: 77%, 51/66; STA: 86%, 175/209) replied that the written inspection report was clear and easy to understand, with some exceptions (CAB: 9%, 6/66; STA: 2%, 4/209). 

## 4. Discussion

### 4.1. Positive and Fair Experience

To our knowledge, this is the first study to investigate trotting horse trainers’ perceptions of animal welfare inspections of their premises. A couple of studies were performed on farmers’ perceptions of animal welfare inspections but focused on farms with conventional production animal species, e.g., [22,23,24,25]. These studies found that farmers, in general, understand the need for animal welfare inspections, since some farmers will be below the standards. Similarly, we found that Swedish trotting horse trainers understand the necessity of being inspected both by the CAB and the STA, although the trainers viewed the STA inspections as more necessary. However, the trotting horse trainers in this study seemed, in general, to have a more positive experience of being inspected than the farmers in previous studies, possibly because they correctly evaluated the standard of their own premises as reasonably high or perhaps because most of the trotting horse trainers had their horses as a hobby and not as their only financial income. Veissier and co-workers [24] found that some French farmers viewed inspections as time consuming, bureaucratic, and not relevant for animal welfare. In our study too, most of the trainers believed that the inspections did not improve actual horse welfare. This perception did not correlate with the trainer’s overall experience of the inspection, e.g., they could be positive about an inspection even if they did not believe that it had improved horse welfare. An explanation for this may be that the trotting horses, in general, were in good condition. Almost no non-compliances reported related to horse welfare (animal-based measures), even though both the CAB and STA inspectors assess horse welfare. The non-compliances were mainly related to the stable and interior design, i.e., risk factors in the horses’ environment. Väärikkälä and co-workers [22] discovered that Finnish farmers who had received an inspection had a more negative attitude towards these inspections, while Anneberg et al. [23] found that Danish farmers perceived the inspections to be generally unfair. In our study, there was no significant difference in perception of the necessity for official inspections between trainers who had received an inspection and those who had not. A majority of the trainers also stated that the inspections had been fair. As expected, those trainers who reported that they had experienced unfairness during an inspection also reported a more negative perception of the inspection.

Having a positive attitude to being inspected can be correlated with a higher level of compliance. Läikkö-Roto and Nevas [26] found that Finnish restaurant business operators with a more positive attitude to official inspections also had fewer non-compliances in food hygiene. In our study, non-compliances were mainly related to the stables, not the actual condition of the horses, and very few harsh sanctions were given to the trainers. The trainers were also generally positive about the inspections, and we found a correlation between satisfaction with the existing regulations, understanding the necessity for these, and how the inspections were perceived.

The trainers were, in general, more positive to the private STA inspections than to the official CAB inspections. There could be several reasons for this. For example, approximately half the trainers surveyed had never received any official inspection from the CAB, while most had received a STA inspection within the previous three years. Hence, it might have been easier to see the necessity of inspections that are carried out more often, since the trainers, in general, had the view that an animal welfare inspection should be carried out at least every third year. A second explanation could be a difference between being inspected by an organization of which the trainer has chosen to be a member and being inspected by an official authority representing the government. A third explanation could be the inspectors’ level of knowledge of the trotting horse business. The CAB inspectors carry out inspections on different types of animal premises, e.g., farms, zoos, research facilities, other horse stables, etc., in addition to trotting horse premises. The STA inspectors only inspect trotting horse trainers and are very specialized in this field. The STA inspectors also all have a background in the industry themselves [17] and can, hence, be perceived as ‘one of us’ by the trainers and as a true expert in the field. Knowledge of horses and of the trotting horse business was the inspector trait that the trainers valued the most, and the trainers perceived the STA inspectors as more competent than the CAB inspectors. There is a consensus within social science that expert knowledge is the main factor for perceiving an inspector as trustworthy [27]. Trustworthiness is important in order to get the message across during an inspection [27]. Other factors that are important for trustworthiness are the presence of unbiased inspectors and a connection or affinity between the inspected individual and the inspector. A fourth explanation could be that the official and private inspectors apply different frameworks. The official inspectors must follow EU regulation 2017/625 on official controls and other national legislations on how to perform inspections. For example, they must not act simply as an advisor, and there are restrictions on pre-announcing inspections. The private inspectors do not necessarily have such a formal framework and might be more open to giving tips and advice. It is also possible that the STA inspectors’ specialization in trotting horse premises facilitates their ability and confidence to give good advice. Our survey indicated that the trainers received more advice from STA than from CAB inspectors, and there was a positive correlation between how the inspection was perceived and the delivery of advice on good practice. Finally, an interesting finding was that the trainers perceived it as more expensive to rectify non-compliances identified by CAB than by STA. Further research is needed to determine whether this is a true observation or a subjective perception. The tools for achieving compliance differ substantially between official inspections and STA audits, as the CAB has an escalating toolbox of remarks and sanctions available in order to force an animal owner to reach compliance [28], while the STA does not have access to a formal, legal toolbox. One possible explanation for the difference could be that the STA leaves more comprehensive non-compliances relating to stable construction for the CAB inspection to handle.

### 4.2. Importance of Inspector Traits

Interestingly, the perceptions of an inspection were not dependent on the outcome, i.e., it did not seem to matter for the perception whether the inspector found non-compliances or not. In contrast, a Finnish study found a correlation between inspection outcome and how the inspection was perceived by farmers [22]. In our study, inspector traits were of greater importance for how the inspection was perceived and were almost always significantly correlated to the perception. Several other studies [22,23,24] emphasized the importance of communication between the inspector and the inspected farmer. Kettunen and co-workers [29] found that the better a food business operator rated the cooperation with an inspector, the higher they rated the quality and benefits of the official inspection. The results in our survey emphasized the importance of the inspector’s knowledge, manner, behavior, and responsiveness. Väärikälä and co-workers [22] found that it is important that inspection reports are clearly written and communicated. In our study, we found a significant correlation between trotting horse trainers’ perception of an STA inspection and the written reports, i.e., if they found new non-compliances in the written report that had not been communicated during the inspection, they perceived the inspection in a more negative way.

### 4.3. Same but Different

The Trotting Health Standard is built upon the Swedish animal welfare legislation, i.e., it mainly consists of exactly the same requirements and is striving for the same animal welfare level as the official legislation. Despite this, more trainers were positive to the Trotting Health Standard, reporting that it is easier to comply with and contains fewer complicated and unnecessary requirements. While the trainers reported that they generally knew how to comply with the legislation and the standard, a large proportion could not answer the more detailed questions relating to the content of the regulations. This may indicate that the trainers are not well informed about similarities and differences between the two regulations and that preconceptions and expectations probably affect their responses. This may also reflect a general perception of activities originating from the government as being complicated and theoretical and activities originating from the industry as being more hands-on, realistic, and reasonable, even when the requirements are, in fact, the same or very similar. This notion is by no means specific to the field of animal welfare.

### 4.4. Limitations of the Study and Dropout Analysis

There are always advantages and disadvantages with a given method. Using online questionnaires has several advantages, e.g., you can access large and geographically distributed populations and achieve quick returns [30]. By making the questionnaire digital, the study also becomes more cost effective and collection, storage, and visualization of data are facilitated [31]. Challenges, on the other hand, are related to, e.g., response rate and non-respondent characteristics [31]. According to Lefever and co-authors [30], online questionnaires may not be as appealing as once believed. In our study, approximately 14% of the trainers who received the link to the online questionnaire responded to some degree. Not all responses were complete. An advantage for us was that we received email lists of the trotting horse trainers and could, thereby, target the response group [30]. The fact that the questionnaire was only digital and quite extensive, and, hence, time consuming, may have contributed to the relatively low number of responses. However, the demographics of the respondents reflected the population of Swedish trotting horse trainers quite well with regard to age and type of license. The overall population of trotting horse trainers in spring 2021 was, on average, 52.3 years of age, and the majority (89%) had a B license [11]. The gender proportion was not reflected to the same extent, with more women (54% of the total) answering the questionnaire, while the overall population of trainers in 2021 consisted of more men (58% men, 42% women). There were only very few situations where gender had a significant effect on the response. However, gender did seem to matter during STA inspections, with female trainers being more positive to these inspections. This may have contributed to the more positive perceptions of STA inspections. Finally, it is important to state that our results cannot be applied on other type of horse owner.

## 5. Conclusions

Trotting horse trainers in Sweden participating in this study were, in general, satisfied with both official CAB control and the animal welfare inspections carried out by the STA. However, they were more satisfied with the STA inspections and saw greater benefit of these inspections. Reasons for this are, e.g., that STA inspections are carried out much more often than official CAB inspections, the STA inspectors specialize in trotting horse activities and premises, and the STA inspectors represent an organization of which the trainer is a member. The outcome of inspections was not associated with trainers’ perception of the inspections but was strongly associated with trainers’ perceptions of the inspector’s knowledge, manner, and responsiveness. The trainers were, in general, satisfied but mentioned that they would like better coordination between the different inspections. Many trainers also said that they would like to see inspections at shorter intervals than currently applied.

## Figures and Tables

**Figure 1 animals-12-01441-f001:**
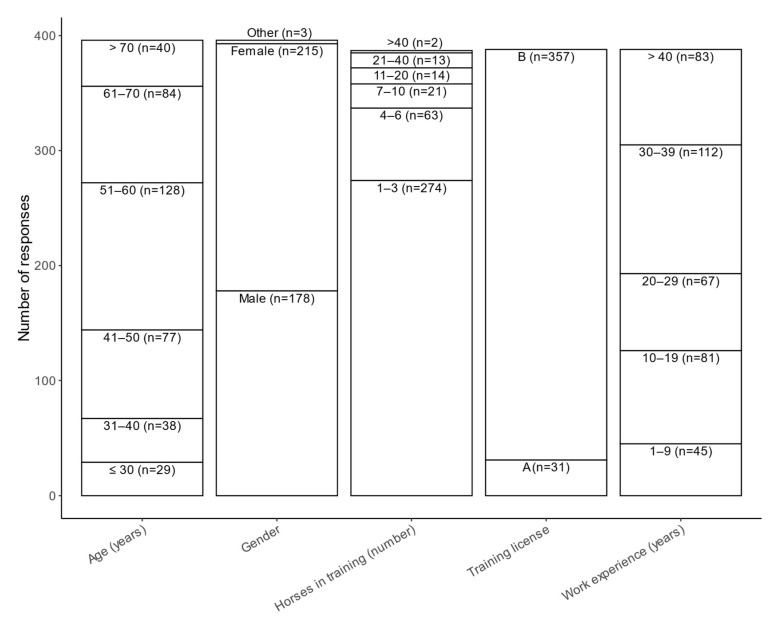
Descriptive information on respondents.

**Figure 2 animals-12-01441-f002:**
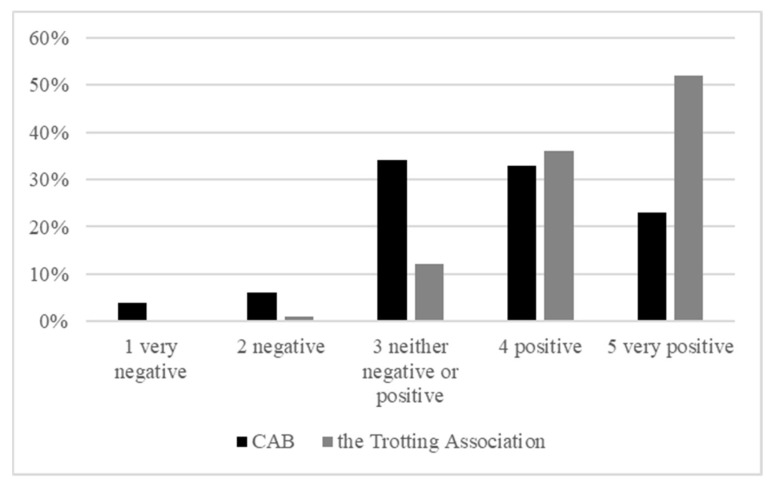
Responses of Swedish trotting horse trainers regarding positive or negative experiences of their latest inspection from the County Administrative Board (CAB) (*n* = 70) or the Swedish Trotting Association (STA) (*n* = 214).

**Table 1 animals-12-01441-t001:** Perceptions (gradings) of respondents who had received both a County Administrative Board (CAB) and Swedish Trotting Association (STA) inspection on these inspections. Cells show number of responses for each combination of grades (1–5). *n* = 54.

Perception of STA		Perception of CAB				
		1	2	3	4	5
	1	0	0	0	0	0
	2	0	0	0	0	0
	3	0	2	4	4	0
	4	1	0	7	12	2
	5	1	0	5	4	12

**Table 2 animals-12-01441-t002:** Results of correlation tests between graded questions and trainers’ perceptions of County Administrative Board (CAB) and Swedish Trotting Association (STA) inspections. For each question, the mean grading is shown for CAB and STA.

		CAB		Correlation with Perceptions of the CAB Inspection			STA		Correlation with Perceptions of the STA Inspection		
	Question/Statement	Mean Response	*n*	*r*	*p*-Value	*n*	Mean Response	*n*	*r*	*p*-Value	*n*
Understanding and expectation ^b^	It is easy to understand what is required in order for me to fulfil the regulation	4.1	326	0.21	0.085	70	4.4	241	**0.30**	<0.001	192
	I am generally satisfied with the regulation	3.8	325	**0.25**	0.040	70	4.3	236	**0.48**	<0.001	188
	[CAB/STA] inspections are needed to ensure that animal welfare is good on Swedish trotting training premises	3.5	303	0.22	0.070	70	4.5	258	**0.38**	<0.001	214
	Were you worried about the [CAB/STA] animal welfare inspection?	8.7	70	0.13	0.285	70	9.3	213	**0.18**	0.008	213
	I am well acquainted with the similarities and differences that exist in terms of requirements and assessments between the legislation and the standard of STA ^a^	3.6	287	0.15	0.249	61	3.6	287	0.08	0.269	183
	I’m happy as it is (having inspections from different actors) ^a^	3.5	294	**0.46**	<0.001	63	3.5	294	**0.25**	0.001	185
Inspector traits and fair treatment ^c^	The inspector was pleasant and had good intentions	4.2	66	**0.62**	<0.001	66	4.7	211	**0.43**	<0.001	211
	The inspector took my opinions and my skills into account	3.8	66	**0.64**	<0.001	66	4.4	210	**0.47**	<0.001	210
	The inspector was knowledgeable and acted professionally	3.8	66	**0.65**	<0.001	66	4.6	210	**0.49**	<0.001	210
	The inspector appeared to be interested in my business	3.7	66	**0.61**	<0.001	66	4.4	210	**0.52**	<0.001	210
	The inspection was fair	3.9	66	**0.52**	<0.001	66	4.6	212	**0.48**	<0.001	212
	The inspection disrupted my routines and those of my business	2.0	66	**−0.39**	0.001	66	1.4	212	**−0.28**	<0.001	212
	I think the inspector made a correct assessment and management of non-compliances	3.4	18	**0.75**	<0.001	18	4.3	74	**0.54**	<0.001	74
	The inspector was confident in their assessment	3.7	18	**0.54**	0.021	18	4.5	74	**0.44**	<0.001	74
	The inspector gave me advice on how I could rectify the non-compliance(s)	3.5	18	**0.48**	0.042	18	4.5	74	**0.49**	<0.001	74
	I was involved and able to influence how long time I had to reach compliance	2.8	17	**0.60**	0.010	17	3.1	72	0.01	0.906	72
Outcome of an inspection ^d^	The inspector explained/justified why something was a deficiency and how it risked affecting the horses’ welfare	3.6	19	**0.55**	0.016	19	4.5	74	**0.51**	<0.001	74
	The inspector had the ability to explain and justify their assessments so that I understood	3.8	66	**0.74**	<0.001	66	4.6	210	**0.48**	<0.001	210
	It was expensive to rectify the non-compliance(s)	2.7	17	**−0.74**	0.001	17	1.6	72	−0.12	0.311	72
	The inspection contributed to better horse keeping and better animal welfare	1.7	66	0.09	0.455	66	2.4	211	0.09	0.176	211

Descriptive statistics for each question are shown with a grey background and grey text color, for each of the two inspections. Correlation coefficients with significant *p*-values are bold. ^a^ A general question, not repeated for each of the two actors/inspections. ^b^ See also Section 3.2, Section 3.3 and Section 3.6.2 for more results on understanding and expectation ^c^ See also Section 3.4 and Section 3.6.3 for more results on inspector traits and fair treatment ^d^ See also Section 3.6.4 for more results on outcome of an inspection.

**Table 3 animals-12-01441-t003:** Results of association tests (Fisher’s exact test) between categorical questions and perceptions of County Administrative Board (CAB) and Swedish Trotting Association (STA) inspections. For each question, each response is shown for CAB and STA.

		CAB	Association with Perceptions of the CAB Inspection			STA	Association with Perceptions of the STA Inspection		
	Question and Responses	Response Distribution	Mean Perception of Inspection	*n*	*p*-Value	Response Distribution	Mean Perception of Inspection	*n*	*p*-Value
Understanding and expectation ^a^	Are there rules in the regulation that you do not consider to benefit the welfare of horses in practice?	*n* = 315		70	**0.017**	*n* = 262		214	**0.012**
	Yes	18%	3.15	13		6%	4.00	13	
	No	23%	4.27	15		51%	4.50	115	
	I do not know	59%	3.57	42		42%	4.29	86	
Inspector traits and fair treatment ^b^	Do you know the reason for the inspection [CAB/STA]?	*n* = 69		69	0.361	*n* = 213		213	0.061
	It was a planned routine inspection	65%	3.73	45		92%	4.41	197	
	It was a follow-up due to previous non-compliances	3%	4.00	2		0%	4.00	1	
	I do not know why I got an inspection	14%	3.10	10		7%	4.07	15	
	Someone had reported to [CAB/STA] that I had deficiencies in my horse keeping	17%	3.58	12					
	What age (in relation to yourself) and gender was the [CAB/STA] inspector(s)?	*n* = 62		62	1.000	*n* = 206		206	0.467
	Only younger women/woman	65%	3.58	40		33%	4.36	69	
	Remaining ages and gender	35%	3.73	22		67%	4.42	137	
	How many inspectors from [CAB/SHTA] attended that inspection?	*n* = 65		65	0.717	*n* = 211		211	0.592
	One	57%	3.65	37		87%	4.39	184	
	Two	43%	3.68	28		12%	4.40	25	
	Three					1%	5.00	2	
	Was the inspection announced beforehand?	*n* = 62		62	0.133				
	Yes	74%	3.83	46					
	No, it was announced the same day or not at all	26%	3.31	16					
Outcome of an inspection ^c^	When you read the inspection report, did new non-compliances emerge that the inspector had not pointed out during the inspection?	*n* = 18		18	0.461	*n* = 73		73	**0.007**
	Yes	17%	2.67	3		5%	3.25	4	
	No	83%	3.73	15		95%	4.42	69	
	Did [CAB/SHTA] find any non-compliances on your premises?	*n* = 58		58	0.400	*n* = 200		200	0.145
	Yes	33%	3.42	19		36%	4.34	73	
	No	67%	3.82	39		64%	4.40	127	

For each question, the responses and results per response are shown with grey text color. Descriptive statistics for each question and response are shown with a grey background color, for each of the two inspections. Significant *p*-values are bold. ^a^ See also Section 3.2, Section 3.3, and Section 3.6.2 for more results on understanding and expectation ^b^ See also Section 3.4 and Section 3.6.3 for more results on inspector traits and fair treatment ^c^ See also Section 3.6.4 for more results on outcome of an inspection.

## Data Availability

The data presented in this study are available on request from the corresponding author.

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
