# Peer review of "Swedish Trotting Horse Trainers’ Perceptions of Animal Welfare Inspections from Public and Private Actors"

_animals, 2022, doi:10.3390/ani12111441_

Round 1

Reviewer 1 Report

The study investigated the perceptions of horse trainers on the inspections carried out by public and private authorities on horse welfare.

The title should be probably more focused: something like that, Harness racing trainers’ perceptions of the inspections for safeguarding horse  welfare in Sweden

Introduction: Please add the major findings of your previous studies, so that the readers do not need to read it.

Add the hypothesis of this study, before the aims.

M&M:

 Line 104 instead of (1 = fully disagree to 5/10 = fully agree), please write  (from 1 = fully disagree to 5 or10 = fully agree)

The majority of the questions are open, and they have been categorised. Please, add how the questions were categorised. This should be presented in a clear table in the M&M section

Statistical Analysis: The questions with the Likert scale should be analysed using ordinal regression analysis. Regression analysis would also be better than with Pearson’s chi-square test or Fisher’s exact test.  

Try to see if there was any difference in the answers between trainers with license A and B (Chi square test).

Results:

It is important to add the sample size needed to be significant for the population investigated, and report the C.I. and margin of error for the obtained sample size 

Currently, they are very difficult to read. The results should be presented in tables or graphs to be more reader-friendly. A test among the category selected within the same question could be done, to see if there is any statistical difference among the percentages. This would add something more to the descriptive statistics.

Discussion

After having improved the statistical analysis the discussion can be improved by interpreting the main results

Please add limitations of the study in the discussion

Author Response

Rev: The title should be probably more focused: something like that, Harness racing trainers’ perceptions of the inspections for safeguarding horse welfare in Sweden
Our response and action: The suggestion from the reviewer introduces terms and concepts that have not been used in the article, which we think would be strange. However, we appreciate the suggestion that the title should be more focused and hence we suggest the new title “Swedish trotting-horse trainers’ perceptions of animal welfare inspections from public and private actors” (see line 3).

Rev: Introduction: Please add the major findings of your previous studies, so that the readers do not need to read it.
Our response and action: We have added a short description of major findings from previous studies. Please see line 76-79.

Rev: Add the hypothesis of this study, before the aims.
Our response and action: The hypothesis has been added. Please see line 81-84.

M&M:

Rev: Line 104 instead of (1 = fully disagree to 5/10 = fully agree), please write  (from 1 = fully disagree to 5 or10 = fully agree)
Our response and action: We have changed according to the reviewers comment. Please see line   113-114.

Rev: The majority of the questions are open, and they have been categorised. Please, add how the questions were categorised. This should be presented in a clear table in the M&M section
Our response and action: The majority of the questions were not open but categorical. However, responses to some categorical questions were regrouped into larger groups to enable statistical analysis. We understand that we might not have explained this clear enough and have therefore added information on lines 119, 123 and 125 to clarify this. Regarding a table showing how the questions were categorized, we suggest adding this table (attached) as supplementary material.

Rev: Statistical Analysis: The questions with the Likert scale should be analysed using ordinal regression analysis. Regression analysis would also be better than with Pearson’s chi-square test or Fisher’s exact test.  
Our response and action: The questionnaire used in this study had many questions but a limited number of responses. We have discussed how to deal with this in the statistical analysis and have chosen to analyze the relationship between question pairs instead of a more complicated model. This is an exploratory study, and we prefer to keep the analysis simple. For example, building a multivariable regression model for each potential outcome would be very complicated and time-consuming. If based on the available data from the questionnaire, a multivariable regression approach would also lead to very few observations for each covariate pattern, which would make estimates less robust. Therefore, this was not the best option for our aims. For the questions pairs with Likert scale we have instead chosen Spearman's correlation to investigate correlation between how the respondents answered the two questions and Pearson’s chi-square test or Fisher’s exact test to investigate potential differences in the response distributions of the two questions. Regression analysis could have been chosen to investigate relationships between these question pairs, but we chose other methods which we thought suitable. As the reviewer has not given a substantial motivation to why we should have chosen regression analysis instead, we suggest keeping the original methods. We did add information in the method-section about why we chose the methods we used, see line 145-147.

Rev: Try to see if there was any difference in the answers between trainers with license A and B (Chi square test).
Our response and action: We have already made this test, with no significant difference between A and B licensed trainers. Please see line 287.

Results:

Rev: It is important to add the sample size needed to be significant for the population investigated, and report the C.I. and margin of error for the obtained sample size 
Our response and action: Since this is not an experimental study but an exploratory questionnaire-based study, our opinion is that sample size calculations are of less value. If these calculations should be included it would also implicate separate calculations for each question. However, we have added some information regarding the limitations of the study design in chapter 4.4. Please see line 532-555.

Rev: Currently, they are very difficult to read. The results should be presented in tables or graphs to be more reader-friendly. A test among the category selected within the same question could be done, to see if there is any statistical difference among the percentages. This would add something more to the descriptive statistics.
Our response and action: We do understand that the results can be perceived as difficult to read, since there is a lot of results. We have struggled with how to report these in the best way without losing too many interesting details for the readers and researchers that are particularly interested in this area of research. This is why we have presented some descriptive results together with results from the statistical analysis in tables, but also included the main findings in the text together with descriptive statistics (if not already presented in the tables). Table 3 displays percentage of responses per category, how respondents in each category graded the perception of the inspection (mean value) and if there was a significant difference in the perception of the inspection based on the chosen category. If we interpret the reviewer's comment correctly, we have already done what is suggested. However, to further help the reader we have made some clarifications in relation to the text and tables. Please see line 298-302. 

Discussion

Rev: After having improved the statistical analysis the discussion can be improved by interpreting the main results
Our response and action: As discussed above, under comment regarding choice of statistical analysis, we prefer to keep the analysis simple and not build more advanced models than the can support. Following our aim to perform this exploratory study, we have highlighted the main results on the basis on the statistical analysis we have used and find suitable (please see comment above regarding choice of statistical analysis). We have included these results in the discussion and interpreted them. Please see for example, chapter 4.1 regarding the trainers’ positive attitude to the inspections in general and more positive perceptions of private inspections compared to official inspections, and chapter 4.2 regarding that inspector traits were more important than the inspection outcomes in relation to how an inspection was perceived. We also prefer to be a little bit careful with our conclusions, in order to avoid speculations.

Rev: Please add limitations of the study in the discussion
Our response and action: We have renamed chapter 4.4 in the discussion to “Limitations of the study and dropout analysis” and added more text regarding limitations (and benefits) with the chosen method. Please see line 532-555.

Other changes:

Numbers in table 2:

In table 2 we show the number of answers (n) for each separate numerical question and the mean value of the grading. However, for some of these graded questions it was also possible to answer “Don’t know” which was removed from the statistical analysis and when calculating the mean value. But the n-values in the table previously included these “Don’t know” answers which has now been corrected. See table 2 (line 412). The following corrections were made: 351 -> 287 (x2), 350 -> 294 (x2) and 18 -> 17.

Number of references:
In order to reach the journals demand of at least 30 references we have added relevant references to the manuscript.

Reviewer 2 Report

animals-1693610 - Swedish trotting-horse trainers’ perceptions of animal welfare inspections   The manuscript addresses a very important issue of the welfare of horses participating in races. Proper control and legal regulations provide a kind of protection for the welfare of these horses. I think an important goal is to check how the formal guidelines are perceived in practice by the trainers. Overall, the manuscript has been prepared with good care, but I have a few questions for the Authors.
  • Did the Authors validate the survey
  • I propose to modify table 1 into figure
  • The numbering in Chapter 3 "Results" made it quite confusing (it's about subchapters)
  • I would like the description of the results contained in subchapters 3.2; 3.3; 3.4 was reflected in the figures

Author Response

Rev: Did the Authors validate the survey

Our response and action: Yes, we did and we have now added information about this in the manuscript. Please see line 94.

Rev: I propose to modify table 1 into figure
Our response and action: We have made a visualization instead and replaced table 1 with this new figure (figure 1). See line 160. This led to changed numbers of the tables and figures which has been adjusted through out the article.

Rev: The numbering in Chapter 3 "Results" made it quite confusing (it's about subchapters). I would like the description of the results contained in subchapters 3.2; 3.3; 3.4 was reflected in the figures

Our response and action:

Firstly, there seems to be duplicated heading numbers (3.3 and 3.4 appeared two times). So, we have changed the number of the headings on lines 269 and 284 to 3.5 and 3.6. Secondly, to facilitate the understanding of how the results in the subheadings and results in tables are related (as the reviewer pointed out) we have added information in the tables on how these results are also reflected in some subchapters. See table 2 and 3 and table footers on lines 414-416 and 421-423.

Other changes:

Numbers in table 2:

In table 2 we show the number of answers (n) for each separate numerical question and the mean value of the grading. However, for some of these graded questions it was also possible to answer “Don’t know” which was removed from the statistical analysis and when calculating the mean value. But the n-values in the table previously included these “Don’t know” answers which has now been corrected. See table 2 (line 412). The following corrections were made: 351 -> 287 (x2), 350 -> 294 (x2) and 18 -> 17.

Number of references:
In order to reach the journals demand of at least 30 references we have added relevant references to the manuscript.